# Impact of T790M Mutation Status on Later-Line Osimertinib Treatment in Non-Small Cell Lung Cancer Patients

**DOI:** 10.3390/cancers14205095

**Published:** 2022-10-18

**Authors:** Yan-Jei Tang, John Wen-Cheng Chang, Ching-Fu Chang, Chen-Yang Huang, Cheng-Ta Yang, Chih-Hsi Scott Kuo, Yueh-Fu Fang, Ping-Chih Hsu, Chiao-En Wu

**Affiliations:** 1Division of Hematology-Oncology, Department of Internal Medicine, Chang Gung Memorial Hospital at Linkou, Chang Gung University College of Medicine, Taoyuan 333, Taiwan; 2Division of Thoracic Oncology, Department of Thoracic Medicine, Chang Gung Memorial Hospital at Linkou, Chang Gung University College of Medicine, Taoyuan 333, Taiwan

**Keywords:** epidermal growth factor receptor, lung cancer, T790M mutation, osimertinib

## Abstract

**Simple Summary:**

Osimertinib is a third-generation epidermal growth factor receptor tyrosine kinase inhibitor (EGFR-TKI) designed to overcome acquired T790M resistance mutations in non-small cell lung cancer (NSCLC). A total of 172 patients with advanced NSCLC treated with osimertinib following frontline EGFR-TKIs were retrospectively reviewed and divided into three groups based on the T790M status (positive, negative, or unknown T790M). The study confirmed the greater efficacy of later-line osimertinib for NSCLC with acquired T790M mutation than for NSCLC without acquired T790M mutation. Detection of the T790M mutation after frontline treatment (first- and second-generation EGFR-TKI) is crucial for prolonging the survival of NSCLC patients harboring EGFR mutation. Osimertinib may be considered an option for NSCLC with unknown T790M mutations, as a certain subpopulation may benefit from osimertinib.

**Abstract:**

Background: Osimertinib is a third-generation epidermal growth factor receptor tyrosine kinase inhibitor (EGFR-TKI) designed to overcome acquired T790M resistance mutations in non-small cell lung cancer (NSCLC). However, the efficacy of osimertinib in patients without acquired T790M mutations has not been well studied. This study aimed to evaluate the efficacy of osimertinib in patients treated with first- and second-generation EGFR-TKIs followed by later-line osimertinib treatment. Patients: The clinical data and survival outcomes of 172 patients with advanced NSCLC treated with osimertinib following frontline EGFR-TKIs at Chang Gung Memorial Hospital from 2014 to 2018 were retrospectively reviewed. T790M mutations were detected using tissue sequencing and/or liquid biopsy. Results: A total of 172 EGFR-mutated NSCLC patients treated with frontline EGFR-TKI therapy followed by later-line osimertinib were enrolled in the current study and divided into three groups based on the T790M status (positive, negative, or unknown T790M). Patients with NSCLC harboring acquired T790M mutation treated with osimertinib had the best objective response rate (ORR) (52.6%, 25.0%, and 32.0%, *p* = 0.044), disease control rate (DCR) (79.3%, 41.7%, and 68.0%, *p* = 0.011), and progression-free survival (PFS, median PFS, 12.6, 3.1, 10.4 months, *p* = 0.001) among the three groups (positive, negative, and unknown T790M, respectively). However, a marked difference was found between positive and negative T790M mutations but not between positive and unknown T790M mutations. Univariate analysis was performed to identify potential prognostic factors for PFS in 172 patients treated with osimertinib. Lung metastasis (*p* < 0.001), brain metastasis (*p* < 0.009), number of metastatic sites (*p* < 0.001), PFS with frontline EGFR-TKIs (*p* = 0.03), and T790M status (*p* = 0.006) were identified as prognostic factors for PFS with osimertinib. Multivariate analysis showed that lung metastasis (*p* < 0.001) and PFS with frontline EGFR-TKIs and T790M status were independent prognostic factors. Conclusion: This study confirmed the greater efficacy of later-line osimertinib for NSCLC with acquired T790M mutation than for NSCLC without acquired T790M mutation. Detection of the T790M mutation after frontline treatment (first- and second-generation EGFR-TKI) is crucial for prolonging the survival of NSCLC patients harboring EGFR mutation. Osimertinib may be considered an option for NSCLC with unknown T790M mutations, as a certain subpopulation may benefit from osimertinib.

## 1. Introduction

In patients with advanced or metastatic non-small cell lung cancer (NSCLC) with mutations in the gene encoding epidermal growth factor receptor (EGFR) that are sensitive to tyrosine kinase inhibitors (TKIs), guidelines recommend treatment with an EGFR-TKI [1,2,3]. The detection of mutations in the kinase domain of the epidermal growth factor receptor provides guidance for advanced non-small cell lung cancer, and EGFR tyrosine kinase inhibitors are the standard first-line therapy [1,2,4,5]. EGFR mutations are the predictive factor for response to first- and second-generation (1G/2G) EGFR-TKIs such as erlotinib, gefitinib, and afatinib, with response rates of approximately 70%. However, most patients develop resistance to these drugs after an effective treatment [6,7,8,9]. The most frequent resistance mechanism is the *p*.Thr790Met point mutation (T790M) of *EGFR*, which is detectable in approximately 60% of patients at the time of progression after frontline EGFR-TKI treatment [10,11,12,13]. 

Osimertinib is an oral third-generation EGFR-TKI that is selective for both EGFR and T790M resistance mutations, with good activity for brain metastasis [14,15]. In patients harboring the T790M mutation after frontline EGFR-TKI treatment, osimertinib showed a higher objective response rate (ORR) and longer progression-free survival (PFS) than platinum-based chemotherapy [16]. The FLAURA study, a phase III trial, showed that osimertinib substantially prolonged PFS and improved response in patients with EGFR T790M advanced non-small cell lung cancer (NSCLC) and progression to prior EGFR-TKI treatment. 

The survival outcomes of osimertinib treatment in patients with advanced NSCLC and progression on prior first- and second-generation EGFR-TKIs with positive, negative, or unknown T790M mutation are not well-known. This study aimed to comprehensively evaluate the impact of T790M mutation on later-line osimertinib and prognostic factors for PFS of osimertinib in advanced NSCLC patients with later-line osimertinib. 

## 2. Materials and Methods

### 2.1. Data Collection

All 172 study patients were obtained from the Chang Gung Research Database [17,18,19,20], which is an integrated database with multi-institutional standardized electronic medical records from all branches of Chang Gung Memorial Hospital (CGMH) in Taiwan, including information from the cancer registry. This study included the clinical data of patients from the cancer registry in the Linkou, Kaohsiung, Keelung, and Chiayi branches of CGMH from 2010 to 2018.

### 2.2. Patients and Their Clinicopathological Features

Patients who were diagnosed with advanced NSCLC with common EGFR mutations (exon 19 deletion or L858R mutation) and who were treated with 1G/2G EGFR-TKI as first-line treatment were enrolled in the study. The details of patients with progression on frontline EGFR-TKIs and subsequent treatment are summarized in Figure 1. Only patients treated with subsequent osimertinib, regardless of T790M status, were enrolled for further analysis.

Clinicopathological features, including age, sex, smoking history, performance status (PS), tumor characteristics, metastatic tumor site, tumor response, and subsequent treatment, were obtained. The patients underwent tissue or liquid biopsy after progression to 1G/2G EGFR-TKI treatment. Both tissue and liquid biopsies showed positive T790M was considered T790M-positive. The majority of tissue biopsies were detected by the arms amplification refractory mutation (ARMS) system, and a minority were detected by next-generation sequencing. The last follow-up period in this study was August 2021.

### 2.3. Tumor Response, Survival, and Statistical Analysis

The tumor response was evaluated based on the Response Evaluation Criteria in Solid Tumors 1.1 criteria, and the detailed definitions of tumor response, PFS, and OS were referred to in our previous study [21]. In this study, PFS and OS from frontline EGFR-TKIs and osimertinib were analyzed, and the results are summarized in Figure 2. 

Continuous variables were compared using ANOVA variance. Categorical variables were compared using Pearson’s chi-square test or Fisher’s exact test based on the expected values. PFS and OS were estimated using the Kaplan–Meier method and compared using the log-rank test. Univariate analysis was performed to evaluate the potential prognostic factors for osimertinib treatment, such as age, sex, PS, smoking history, histology, and location of metastases. Multivariate analysis was performed to identify the independent prognostic factors. The results are presented as hazard ratios (HR) and 95% confidence intervals (CI) according to Cox regression analyses. IBM SPSS Statistics for Windows (version 23.0, Armonk, NY, USA) was used to perform all statistical analyses, and statistical significance was set at *p* < 0.05. Survival curves were plotted using SPSS software. 

### 2.4. Ethical Issue

This study was approved by the Institutional Review Board (IRB) of the CGMH (IRB No. 201901395B0C501). Patient consent to participate was not required because of the retrospective nature of the study.

## 3. Results

### 3.1. Sequential Treatment for Patients after Frontline EGFR-TKI Treatment

In this study, a total number of 2190 EGFR-mutated NSCLC patients were treated with frontline EGFR-TKI therapy, and 172 patients were treated with later-line osimertinib in the current study. Based on the T790M mutation status, patients were divided into three subgroups: subsequent treatment with osimertinib for tumors with positive T790M mutation, negative T790M mutation, and unknown T790M mutation, which accounted for 135, 12, and 25 patients, respectively. No significant differences in potential confounding factors, including baseline characteristics, tumor characteristics, frontline TKI therapy, and metastatic site, were found among the three subgroups (Table 1). The classification of subsequent treatment and the detailed number of patients based on the T790M status are summarized in Figure 1.

### 3.2. Tumor Base Response of Osimertinib Based on T790M Status

Among the 172 patients, the best tumor response of the NSCLC patients treated with later-line osimertinib based on T790M status showed that three (25%) patients had a partial response (PR), and two (16.7%) patients had stable disease (SD) among 12 patients with NSCLC harboring negative T790M; 135 enrolled patients were T790M-positive, including complete responses (CR) in two (1.5%) patients, PRs in 69 (51.1%) patients, SDs in 25 (18.5%) patients, and PDs in three (2.2%) patients; 25 enrolled patients were T790M-unknown, including PRs in eight (32%), SDs in six (8.0%), and PDs in two (8.0%). Moreover, NSCLC patients harboring positive T790M mutations had the highest ORR (52.6% for positive T790M, 25% for negative T790M, and 8% for unknown T790M, *p* = 0.044) and DCR (79.3% for positive T790M, 41.7% for negative T790M, and 68% for unknown T790M, *p* = 0.011). The best tumor responses are summarized in Table 2.

### 3.3. Progression-Free Survival and Overall Survival Based on T790M Status

Progression-free survival (PFS) of frontline EGFR-TKI shows no significant difference from the three groups (positive vs. negative vs. unknown T790M: median PFS: 16.7 vs. 10.1 vs. 16.2 months, log-rank *p* = 0.863; Figure 3). The patients with NSCLC harboring no T790M mutation showed shorter PFS than those with T790M mutation (median PFS: 3.1 vs. 12.6 months, hazard ratio (HR): 2.65, 95% confidence interval (CI): 1.40–5.0, *p* = 0.003). No significant difference was found between the unknown T790M and positive T790M groups (median PFS 10.4 vs. 12.6 months, HR: 1.30, 95% CI: 0.81–2.06, *p* = 0.275) (Figure 4). OS from frontline EGFR-TKIs based on T790M status showed no significant difference between the three groups (positive vs. negative vs. unknown T790M, median OS: 58.3 vs. 38.1 vs. 46.0 months, log-rank *p* = 0.297) (Figure 5). A shorter OS from the initiation of osimertinib was found between the negative T790M group and positive T790M (median month 7.3, 25.3 months, HR: 2.08, 95% CI: 1.04–4.03, *p* = 0.031); otherwise, no significant difference was found between the unknown and positive T790M (median month: 18.0 vs. 25.3months, HR: 1.24, 95% CI: 0.74–2.08, *p* = 0.417) (Figure 6).

### 3.4. Univariate and Multivariate Analyses of Prognostic Factors for PFS after Osimertinib Treatment

Univariate analysis was performed to identify potential prognostic factors for PFS in 172 patients treated with osimertinib. Lung metastasis (with and without lung metastasis, median PFS 7.8 and 18.2 months, *p* < 0.001), brain metastasis (with and without brain metastasis, median PFS 7.8 and 13.3 months, *p* < 0.009), number of meta sites (0–1, 2–3 and ≥4, median PFS 16.9, 8.9 and 8.6 months, *p* < 0.001), PFS of frontline EGFR-TKIs (≤6, 6–12, 12–18 and >24 months, median PFS 3.3, 10.3, 11.3, 9.9 and 22.0 months, *p* = 0.03) and T790M status (negative, positive, unknown, median PFS 3.1, 12.6, 10.4 months, *p* = 0.006) were identified as prognostic factors for later-line osimertinib. Different 1G/2G EGFR-TKI choices were not associated with later-line osimertinib treatment (afatinib, erlotinib, gefitinib, median PFS 11.3, 11.1, 13.4 months, *p* = 0.446). 

Because the number of metastatic sites was related to individual metastasis, only lung metastasis and brain metastasis were included in the multivariate analysis, which showed that lung metastasis (HR: 2.06, 95% CI: 1.41–3.01, *p* < 0.001), PFS of frontline EGFR-TKIs (HR: 2.69, 2.18, 1.96 CI: 1.33–5.44, 1.22–3.90, 1.01–3.79, *p* = 0.006, 0.009, 0.045, ≤6, 6–12, 18–24 months respectively), T790M negative (HR: 2.12, CI: 1.08–4.15, *p* = 0.029) were independent prognostic factors (Table 3).

## 4. Discussion

In this study, NSCLC patients harboring positive T790M mutations experienced better ORR and DCR than those with negative T790M mutations. Patients with NSCLC positive for T790M treated with osimertinib experienced the best PFS and OS after osimertinib treatment. In addition, lung metastasis, brain metastasis, number of metastatic sites, PFS with frontline TKIs, and T790M status were prognostic factors for PFS with osimertinib by univariate analysis. Lung metastasis and PFS with frontline EGFR-TKIs and T790M status were independent prognostic factors in multivariate analysis. Furthermore, patients with unknown T790M mutations did not have significantly worse outcomes than those with positive T790M mutations. These findings suggest that later-line osimertinib should be prescribed to patients with acquired T790M mutations rather than to those without acquired T790M mutations. For patients with unknown T790M mutations, osimertinib is still an option for NSCLC patients with progression on frontline EGFR-TKIs.

In Taiwan, 1G/2G EGFR-TKIs, gefitinib, erlotinib, and afatinib are all commonly prescribed as frontline therapy for EGFR-mutated NSCLC patients [22,23]. Although a previous study reported that frontline EGFR-TKIs (gefitinib, erlotinib, and afatinib) had a statistically significant difference in the T790M mutation rate (59.9%, 45.5%, and 52.7%, respectively; *p* = 0.037) [24], there was no significant difference in the T790M mutation rate after frontline treatment with different EGFR-TKIs. In addition, frontline EGFR-TKIs did not influence the activity of osimertinib in this study, which is different from a previous report [25]; therefore, more studies are warranted to validate this finding. In this study, the T790M mutation significantly influenced the ORR, DCR, and PFS after osimertinib treatment, suggesting that the T790M mutation is a key prognostic factor for osimertinib treatment. Re-biopsy for detecting acquired T790M status is important in NSCLC patients after progression to frontline EGFR-TKI therapy. 

In the present study, not all patients with positive T790M mutations responded to osimertinib treatment. A previous study showed that the ratio of T790M to EGFR-sensitizing mutation allele frequency (AF) influences the efficacy of osimertinib and can predict its response [26]. Moreover, MET/HER2 amplification and RAS-mitogen-activated protein kinase can be other resistance mechanisms to osimertinib in NSCLC patients with positive T790M [27]; therefore, comprehensive genomic testing for these resistance mechanisms should be performed, and adequate treatment should be provided to achieve better survival outcomes.

In the current study, a small population (25%) of patients with negative T790M mutations still responded to osimertinib. This may result from false negatives of genetic testing because of tumor heterogeneity in tissue biopsy and the low sensitivity of liquid biopsy. In a previous study, 36.8% of the 36 patients with negative T790M upon the first biopsy showed positive T790M mutation in the second re-biopsy [28], indicating the benefit of repeated biopsy. In addition, comprehensive genetic testing with a combination of tissue and liquid biopsies can enrich the positive rate for detecting the T790M mutation [17]. 

In patients with unknown T790M mutations, it is estimated that approximately 60% of patients have the T790M mutation based on previous studies [24]. In the current study, although there was no significant difference between the T790M-positive and T790M-unknown groups, a rapid decline at the beginning of the survival curve (Figure 4) in the T790M-unknown group was found, implying that the T790M-negative subpopulation did not respond to osimertinib. For patients without response to osimertinib in the unknown T790M group, patients may experience PD within 2–3 months. Therefore, close monitoring of tumor status is critical for patients with unknown T790M mutations treated with osimertinib to prevent rapid progression and may switch to chemotherapy as soon as possible once progression is documented.

Brain metastasis was a poor prognostic factor for NSCLC patients in this study. Osimertinib showed better ORR than chemotherapy in the phase III AURA3 study (intracranial ORR: 70% with osimertinib vs. 31% with platinum plus pemetrexed in patients with previously treated T790M positive NSCLC, *p* = 0.015) and 1G EGFR-TKIs in a phase III FLAURA study (intracranial ORR: 66% with osimertinib versus 43% with standard EGFR-TKIs in patients with untreated EGFR-mutated NCLC, *p* = 0.011) [29]. Both trials indicated that osimertinib has great intracranial activity in patients with untreated or previously treated EGFR-mutated NSCLC patients. Although brain metastasis was a poor prognostic factor in this study, osimertinib should still be the best option for patients with brain metastasis, particularly in patients with acquired T790M mutations after progression in NSCLC patients. Future studies should investigate novel treatments for NSCLC patients with brain metastasis.

This study had some limitations, the first of which was the small number of negative and unknown T790M patients enrolled in this retrospective study, which may have underestimated the importance of T790M. Survival bias was inevitable in this study. Second, patients underwent tissue biopsy, liquid biopsy, or both, which may have influenced the accuracy of T790M detection. Because of the limited number of cases of negative and unknown T790M, it is not powerful enough to distinguish the significance of using different detection methods. Third, we focused on osimertinib treatment without evaluating the impact of a subsequent immune checkpoint inhibitor, which is also an important treatment for patients with EGFR-mutated NSCLC [30]. However, the current evidence is still limited to EGFR-mutated NSCLC after TKI progression. 

## 5. Conclusions

This study confirmed the greater efficacy of later-line osimertinib for NSCLC with T790M mutation than for NSCLC without T790M mutation. Detection of the T790M mutation after frontline treatment (first- and second-generation EGFR-TKIs) is crucial for prolonging the survival of NSCLC patients harboring EGFR mutations. Osimertinib may be considered for NSCLC with unknown T790M status as a subpopulation that may benefit from osimertinib treatment.

## Figures and Tables

**Figure 1 cancers-14-05095-f001:**
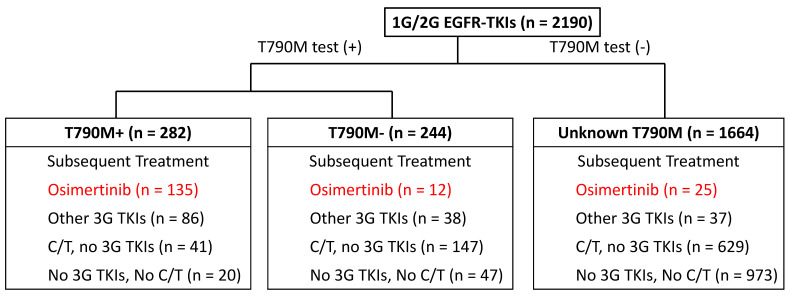
The summary of subsequent treatment following afatinib-based T790M mutation.

**Figure 2 cancers-14-05095-f002:**
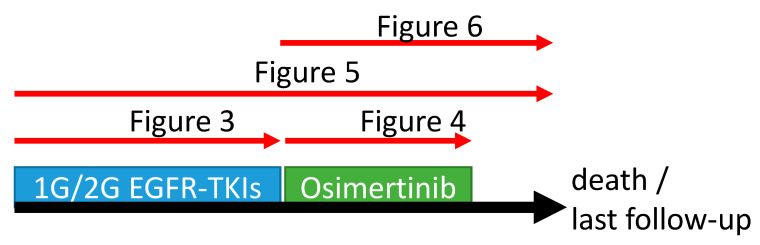
Event-free survival evaluated in the current study.

**Figure 3 cancers-14-05095-f003:**
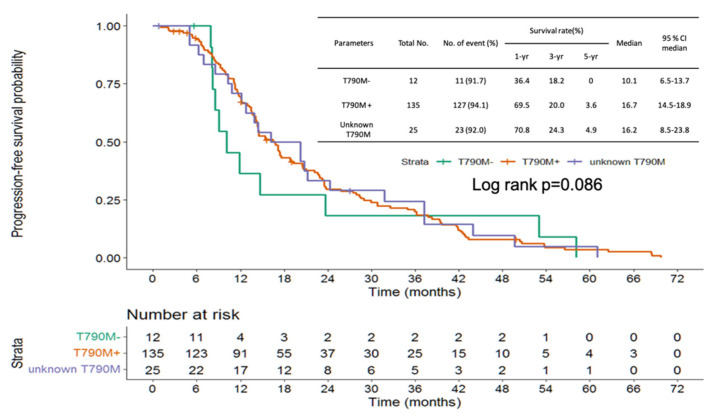
Progression-free survival of frontline EGFR-TKIs based on T790M status.

**Figure 4 cancers-14-05095-f004:**
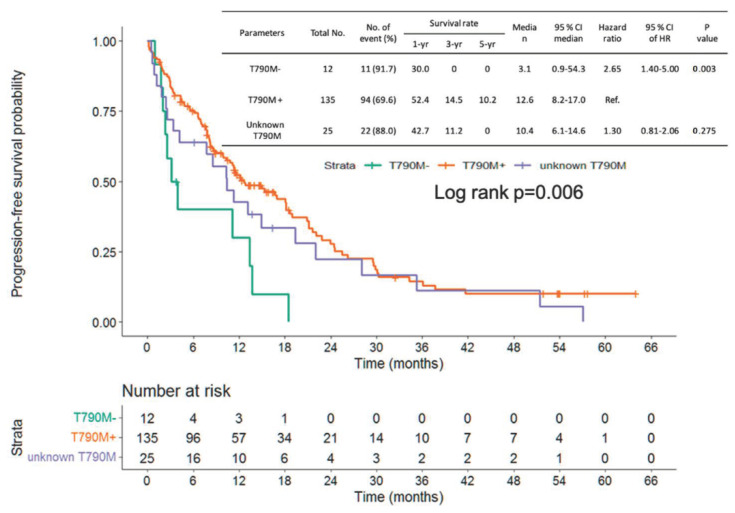
Progression-free survival of osimertinib based on T790M status.

**Figure 5 cancers-14-05095-f005:**
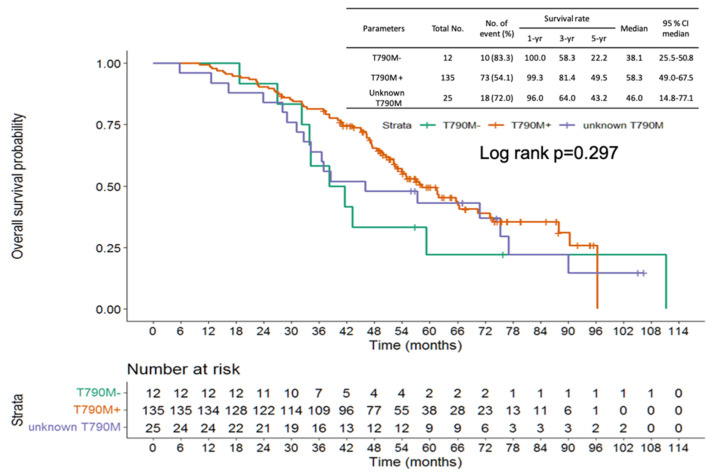
Overall survivals from frontline EGFR-TKIs based on T790M status.

**Figure 6 cancers-14-05095-f006:**
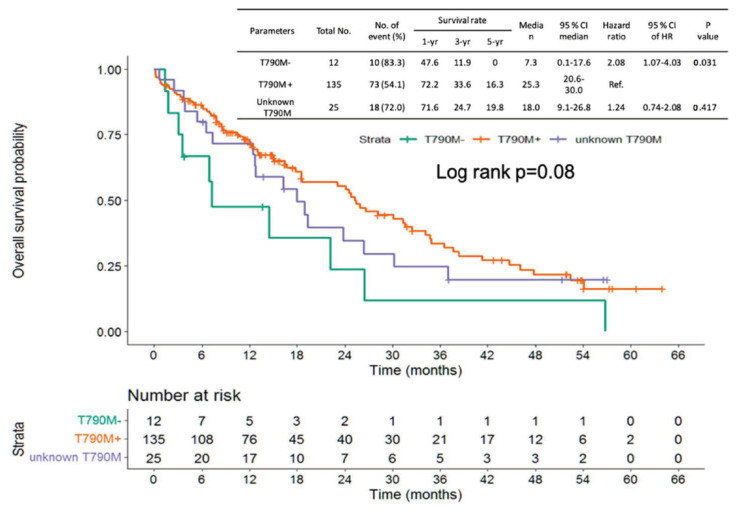
Overall survivals from osimertinib based on T790M status.

**Table 1 cancers-14-05095-t001:** Univariate and Multivariate analysis (*n* = 172).

Factors	Number of Patients	T790M			
		Positive	Negative	Unknown	*p*-Value
*Basic data*					
Age (years)					0.471
≤65	98	74 (75.5)	7 (7.1)	17 (17.3)	
>65	74	61 (82.4)	5 (6.8)	8 (10.8)	
Gender					0.877
Male	76	61 (80.3)	5 (6.6)	10 (13.2)	
Female	96	74 (77.1)	7 (7.3)	15 (15.6)	
Performance score					0.472
0	43	30 (69.8)	3 (7.0)	10 (23.3)	
1	119	97 (81.5)	8 (6.7)	14 (11.8)	
2	8	6 (75.0)	1 (12.5)	1 (12.5)	
3	0	0	0	0	
4	2	2 (100.0)	0	0	
Performance score					0.348
0	43	30 (69.8)	3 (7.0)	10 (23.3)	
1	119	97 (81.5)	8 (6.7)	14 (11.8)	
2/3/4	10	8 (80.0)	1 (10.0)	1 (10.0)	
Smoking					0.829
No	130	99 (76.2)	10 (7.7)	21 (16.2)	
Yes	38	32 (84.2)	2 (5.3)	4 (10.5)	
Unknown	4	4 (100.0)	0	0	
*Tumor characteristics*					
Morphology					>0.999
Adenocarcinoma	171	134 (78.4)	12 (7.0)	25 (14.6)	
Non-adenocarcinoma	1	1 (100.0)	0	0	
Mutation					0.249
19del	108	81 (75.0)	10 (9.3)	17 (15.7)	
L858R	64	54 (84.4)	2 (3.1)	8 (12.5)	
Stage					0.374
IIIB	16	11 (68.8)	2 (12.5)	3 (18.8)	
IV	156	124 (79.5)	10 (6.4)	22 (14.1)	
*TKI therapy*					
Afatinib	69	54 (78.3)	4 (5.8)	11 (15.9)	0.392
Erlotinib	47	41 (87.2)	2 (4.3)	4 (8.5)	
Gefitinib	56	40 (71.4)	6 (10.7)	10 (17.9)	
TTD (months)					0.475
<6	16	12 (75.0)	1 (6.3)	3 (18.8)	
6–12	44	32 (72.7)	7 (15.9)	5 (11.4)	
12–18	42	36 (85.7)	1 (2.4)	5 (11.9)	
18–24	23	18 (78.3)	1 (4.3)	4 (17.4)	
>24	47	37 (78.7)	2 (4.3)	8 (17.0)	
*Metastatic site*					
Lung					0.123
Yes	70	54 (77.1)	8 (11.4)	8 (11.4)	
No	102	81 (79.4)	4 (3.9)	17 (16.7)	
Liver					0.878
Yes	13	11 (84.6)	0	2 (15.4)	
No	159	124 (78.0)	12 (7.5)	23 (14.5)	
Brain					0.167
Yes	43	31 (72.1)	2 (4.7)	10 (23.3)	
No	129	104 (80.6)	10 (7.8)	15 (11.6)	
Bone					0.885
Yes	76	61 (80.3)	5 (6.6)	10 (13.2)	
No	96	74 (77.1)	7 (7.3)	15 (15.6)	
Pleura					0.880
Yes	66	53 (80.3)	4 (6.1)	9 (13.6)	
No	106	82 (77.4)	8 (7.5)	16 (15.1)	
Adrenal					0.422
Yes	9	6 (66.7)	1 (11.1)	2 (22.2)	
No	163	129 (79.1)	11 (6.7)	23 (14.1)	
Distant lymph node					0.860
Yes	11	9 (81.8)	1 (9.1)	1 (9.1)	
No	161	126 (78.3)	11 (6.8)	24 (14.9)	
Pericardia					>0.999
Yes	1	1 (100.0)	0	0	
No	171	134 (78.4)	12 (7.0)	25 (14.6)	
Peritoneum					>0.999
Yes	3	3 (100.0)	0	0	
No	169	132 (78.1)	12 (7.1)	25 (14.8)	
No. of metastatic sites					0.379
0–1	84	67 (79.8)	4 (4.8)	13 (15.5)	
2–3	77	60 (77.9)	8 (10.4)	9 (11.7)	
Four or more	11	8 (72.7)	0	3 (27.3)	

Abbreviations: TKI, tyrosine kinase inhibitor; TTD, time-to-treatment discontinuation.

**Table 2 cancers-14-05095-t002:** Tumor base response of osimertinib based on T790M status.

Response	Total (*n* = 69)	T790M			*p* Value
		Negative (*n* = 12)	Positive (*n* = 135)	Unknown (*n* = 25)	
CR	2 (1.2)	0	2 (1.5)	0	0.051
PR	80 (46.5)	3 (25.0)	69 (51.1)	8 (32.0)	
SD	47 (27.3)	2 (16.7)	36 (26.7)	9 (36.0)	
PD	38 (22.1)	7 (58.3)	25 (18.5)	6 (24.0)	
NA	5 (2.9)	0	3 (2.2)	2 (8.0)	
ORR					0.044
CR/PR	82 (47.7)	3 (25.0)	71 (52.6)	8 (32.0)	
DCR					0.011
CR/PR/SD	129 (75.0)	5 (41.7)	107 (79.3)	17 (68.0)	

Abbreviations: CR, complete response; PR, partial response; SD, stable disease; PD, progressive disease; NA, no assessed; ORR, objective response rate; DCR, disease control rate.

**Table 3 cancers-14-05095-t003:** Univariate and multivariate analysis of prognostic factors for PFS of osimertinib in patients treated with osimertinib.

Parameters	*n* Total	*N* of Events (%)	Median (Months)	95% CI	*p* Value	Hazard Ratio	95% CI	*p* Value
Age (years)					0.197	–		
≤65	98	77 (78.6)	10.3	6.7–13.9				
>65	74	50 (67.6)	13.2	9.1–17.3				
Gender					0.319	–		
Male	76	56 (73.7)	10.3	7.3–13.4				
Female	96	71 (74.0)	12.8	9.3–16.4				
Performance score					0.201	–		
0	43	31 (72.1)	9.9	5.8–14.0				
1	119	88 (73.9)	13.2	10.2–16.2				
2/3/4	10	8 (80.0)	8.2	0.4–16.0				
Smoking					0.528	–		
Yes	38	30 (78.9)	7.6	2.9–12.4				
No	130	95 (73.1)	12.8	9.5–16.2				
Unknown	4	2 (50.0)	11.5	–				
Morphology					0.783	–		
Adenocarcinoma	171	126 (73.7)	11.5	9.0–14.1				
Non-adenocarcinoma	1	1 (100.0)	12.8	–				
Mutation					0.422	–		
19del	108	81 (75.0)	11.3	8.7–14.0				
L858R	64	46 (71.9)	12.6	8.4–16.9				
Stage					0.425	–		
IIIB	16	11 (68.7)	14.9	5.3–24.6				
IV	156	116 (74.4)	11.3	9.2–13.5				
Lung metastasis					<0.0001			
Yes	70	60 (85.7)	7.8	5.7–9.9		2.06	1.41–3.01	<0.001
No	102	67 (65.7)	18.2	11.2–25.1		1		
Liver metastasis					0.498	–		
Yes	13	10 (66.9)	8.9	4.3–13.6				
No	159	117 (73.6)	12.2	9.2–15.1				
Brain metastasis					0.009			
Yes	43	34 (79.1)	7.8	4.2–11.3		1.53	0.99–2.37	0.056
No	129	93 (72.1)	13.2	10.1–16.2		1		
Bone metastasis					0.220	–		
Yes	76	59 (77.6)	11.5	9.0–14.0				
No	96	68 (70.8)	12.2	7.2–17.1				
Pleura metastasis					0.666	–		
Yes	66	53 (80.3)	10.8	8.0–13.5				
No	106	74 (69.8)	13.7	8.6–18.8				
Adrenal metastasis					0.811	–		
Yes	9	8 (88.9)	15.0	0.1–34.5				
No	163	116 (73.0)	11.5	9.3–13.8				
Distant lymph node metastasis					0.491	–		
Yes	11	7 (63.6)	11.3	3.2–19.5				
No	161	120 (74.5)	11.6	9.0–14.1				
Pericardia metastasis					0.406	–		
Yes	1	1 (100.0)	7.8	–				
No	171	126 (73.7)	11.6	9.1–14.1				
Peritoneum metastasis					0.180	–		
Yes	3	2 (66.7)	34.3	0.1–83.8				
No	169	125 (74.0)	11.5	9.3–13.8				
No. of metastatic sites					0.001	–		
0–1	84	55 (65.5)	16.9	8.6–25.2				
2–3	77	62 (80.5)	8.9	5.7–12.1				
Four or more	11	10 (90.9)	8.6	4.0–13.2				
TKI drug					0.446	–		
Afatinib	69	45 (65.2)	11.3	7.9–14.7				
Erlotinib	47	37 (78.7)	11.1	6.1–16.1				
Gefitinib	56	45 (80.4)	13.4	8.6–18.2				
Response					0.459	–		
CR/PR	150	109 (72.7)	12.2	10.1–14.2				
SD	17	13 (76.5)	8.4	0.1–18.4				
PD/NA	5	5 (100.0)	3.3	0.1–8.6				
PFS (months)					0.003			
<6	16	16 (100.0)	3.3	0.1–7.1		2.69	1.33–5.44	0.006
6–12	44	41 (93.2)	10.3	6.7–13.9		2.18	1.22–3.90	0.009
12–18	42	31 (73.8)	11.3	7.5–15.0		1.73	0.96–3.11	0.069
18–24	23	20 (87.0)	9.9	3.7–16.1		1.96	1.01–3.79	0.045
>24	47	19 (40.4)	22.0	15.2–28.9		1		
T790M					0.006			
Negative	12	11 (91.7)	3.1	0.9–5.3		2.12	1.08–4.15	0.029
Positive	135	94 (69.6)	12.6	8.2–17.0		1		
Unknown	25	22 (88.0)	10.4	6.1–14.6		1.63	1.00–2.68	0.051

Abbreviations: CI, confidence interval; TKI, tyrosine kinase inhibitor; CR, complete response; PR, partial response; SD, stable disease; PD, progressive disease; NA, no assessed; PFS, progression-free survival.

## Data Availability

Data presented in this study are available on request from the corresponding author.

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
