# Peer review of "Impact of T790M Mutation Status on Later-Line Osimertinib Treatment in Non-Small Cell Lung Cancer Patients"

_cancers, 2022, doi:10.3390/cancers14205095_

Round 1

Reviewer 1 Report

Reviewer

Initial comments

           This work is very important, because it is a disease that unfortunately affects a large population of people and an effective treatment that can improve people's quality of life and greater survival is essential.

 Abstract

Patients: The clinical data and survival outcomes of patients with advanced NSCLC treated with osimertinib following frontline EGFR-TKIs at Chang Gung Memorial Hospital from 2014 to 2018 were retrospectively reviewed. T790M mutations were detected using tissue sequencing and/or liquid biopsy.

Comment:

Please, could you include the number of patients here.

1. Introduction  

 Comment:

It is suitable

2. Materials and methods  

2.1. Data collection

All study patients were obtained from the Chang Gung Research Database [16-19], which is an integrated database with multi-institutional standardized electronic medical records from all branches of Chang Gung Memorial Hospital (CGMH) in Taiwan, including information from the cancer registry. This study included the clinical data of patients from the cancer registry in the Linkou, Kaohsiung, Keelung, and Chiayi branches of CGMH from 2010 to 2018.

Comment:

Please, how many patients were observed?

2.2. Patients and their clinicopathological features

Comment:

It is suitable

2.3. Tumor response, survivals, and statistical analysis

Comment:

It is suitable

2.3. Ethical issue

Comment:

It is suitable

3. Results

3.1. Sequential treatment for patients after front line EGFR-TKI treatment

Comment:

It is suitable

3.2. Tumor base response of osimertinib based on T790M status

Comment:

It is suitable

3.3. Progression-free survival of treatment and overall survive based on T790M status

Comment:

It is suitable

3.4. Univariate and multivariate analyses of prognostic factors for PFS after osimertinib treatment

Comment:

It is suitable

4. Discussion

Comment:

It is suitable

5. Conclusion

Comment:

It is suitable

References

Comment:

It is suitable

Thank you

Author Response

Patients: The clinical data and survival outcomes of 172 patients with advanced NSCLC treated with osimertinib following frontline EGFR-TKIs at Chang Gung Memorial Hospital from 2014 to 2018 were retrospectively reviewed. T790M mutations were detected using tissue sequencing and/or liquid biopsy.

All 172 study patients were obtained from the Chang Gung Research Database [16-19], which is an integrated database with multi-institutional standardized electronic medical records from all branches of Chang Gung Memorial Hospital (CGMH) in Taiwan, including information from the cancer registry. This study included the clinical data of patients from the cancer registry in the Linkou, Kaohsiung, Keelung, and Chiayi branches of CGMH from 2010 to 2018.

Thank you

Reviewer 2 Report

Dear Authors,

Overall great study and appropriate methods are used to address the question. here are a few suggestions:

1) Introduction can be improved and more content related to NSCLC can be added and slowly progressed towards different treatments 2) please list the full form for all the abbreviations (a small table listing all abbreviations will be very useful for the reader) 3) Please check all the sentences for errors. Thanks for your understanding and co-operation.

Author Response

1.  improved my introduction as following  In patients with advance or metastatic non-small cell lung cancer (NSCLC) with mutations in the gene encoding epidermal growth factor receptor (EGFR) that are sensitive to tyrosine kinase inhibitors (TKIs), guidelines recommend treatment with an EGFR-TKI.[1-3] The detection of mutations in the kinase domain of the epidermal growth factor receptor provides guidance for advanced non-small cell lung cancer, and EGFR tyrosine kinase inhibitors are the standard first-line therapy [1,2,4,5]   2. I checked all the sentences errors  3.
abbreviations
epidermal growth factor receptor-tyrosine kinase inhibitor (EGFR-TKI) non-small cell lung cancer (NSCLC) Chang Gung Memorial Hospital (CGMH) objective response rate(ORR)disease control rate(DCR) progression-free survival (PFS) first- and second-generation(1G/2G) hazard ratios (HR), confidence intervals (CI) complete response (CR), partial response (PR) stable disease (SD),progress disease (PD)

Thank you

Reviewer 3 Report

The manuscript by Tang et al documents the retrospective study of EGFR-TKI positive patients that progressed via T790M mutations.  The study reinforces other reports findings that patients with progressive brain metastases on first- or second-generation EGFR-TKIs can benefit from subsequent osimertinib therapy.  Due to low samples numbers and the lack of detection sensitivity of ARMS, it is difficult to understand the implication of T790M unknown status at this point.  It would be interesting to include the patients which were assessed with next-gen sequencing vs ARMS to identify confounders. 

Also, it would be beneficial to the readers to understand the implication of patients with plasma T790M, especially since those that are negative should be allowed to receive osimertinib treatment.

Minor point:  Figure 2 can be deleted. 

Minor point:  Did the authors find any difference between response to osimertinib based on the choice of front-line TKI?

Minor point:  Line 212, genomic testing does not account for phenotypic transformation.

Author Response

1. I will deleted the figure 2

2. Front-line EGFR-TKIs did not influence the activity of osimertinib in this study. We did not find any difference between response to osimertinib based on the choice of front-line TKI.

3. I changed this sentence to 

Moreover, MET/HER2 amplification and RAS-mitogen-activated protein kinase can be other resistance mechanisms to osimertinib in NSCLC patients with positive T790M[27]; therefore, comprehensive genomic testing for these resistance mechanisms should be performed, and adequate treatment should be provided to achieve better survival outcomes.   This sentence is emphasis on the genomic testing. Although phenotypic transformation is another important factor.    Thank you